# *Casitas B-lineage lymphoma* Gene Mutation Ocular Phenotype

**DOI:** 10.3390/ijms23147868

**Published:** 2022-07-17

**Authors:** Christine Fardeau, Munirah Alafaleq, Marie-Adélaïde Ferchaud, Miguel Hié, Caroline Besnard, Sonia Meynier, Frédéric Rieux-Laucat, Damien Roos-Weil, Fleur Cohen, Isabelle Meunier

**Affiliations:** 1Department of Ophthalmology, Reference Center for Rare Diseases, La Pitié-Salpêtrière Hospital, Paris-Sorbonne University, 47-83 Boulevard de l’Hôpital, 75013 Paris, France; monira-afaleq@hotmail.com (M.A.); maferchaud@gmail.com (M.-A.F.); 2Department of Ophthalmology, Imam Abdulrahman Bin Faisal University, Dammam 34212, Saudi Arabia; 3Department of Internal Medicine, La Pitié-Salpêtrière Hospital, Paris-Sorbonne University, 47-83 Boulevard de l’Hôpital, 75013 Paris, France; miguel.hie@aphp.fr (M.H.); fleur.cohen@aphp.fr (F.C.); 4Laboratory of Immunogenetics of Autoimmune Diseases in Children, INSERM UMR 1163, Imagine Institute, 24 Boulevard du Montparnasse, 75015 Paris, France; caroline.besnard@inserm.fr (C.B.); sonia.meynier@gmail.com (S.M.); frederic.rieux-laucat@inserm.fr (F.R.-L.); 5Hematology Department, La Pitié-Salpêtrière Hospital, Paris-Sorbonne University, 47-83 Boulevard de l’Hôpital, 75013 Paris, France; damien.roosweil@aphp.fr; 6Department of Ophthalmology, Reference Centre for Genetic Sensory Diseases, Hôpital Gui de Chauliac, Montpellier University, 34295 Montpellier, France; i-meunier@chu-montpellier.fr

**Keywords:** macular edema, retinal dystrophy, uveitis, pseudo uveitis, mimicking uveitis, retinal vasculitis, *CBL* gene, RASopathy, papillary edema, *Casitas B-lineage lymphoma*

## Abstract

This article describes the ocular phenotype associated with the identified *Casitas B-lineage lymphoma* (*CBL*) gene mutation and reviews the current literature. This work also includes the longitudinal follow-up of five unrelated cases of unexplained fundus lesions with visual loss associated with a history of hepatosplenomegaly. Wide repeated workup was made to rule out infections, inflammatory diseases, and lysosomal diseases. No variants in genes associated with retinitis pigmentosa, cone–rod dystrophy, and inherited optic neuropathy were found. Molecular analysis was made using next-generation sequencing (NGS) and whole-exome sequencing (WES). The results included two cases sharing ophthalmological signs including chronic macular edema, vascular leakage, visual field narrowing, and electroretinography alteration. Two other cases showed damage to the optic nerve head and a fifth young patient exhibited bilateral complicated vitreoretinal traction and carried a heterozygous mutation in the *CBL* gene associated with a mutation in the *IKAROS* gene. Ruxolitinib as a treatment for RASopathy did not improve eye conditions, whereas systemic lesions were resolved in one patient. Mutations in the *CBL* gene were found in all five cases. In conclusion, a detailed description may pave the way for the *CBL* mutation ocular phenotype. Genetic analysis using whole-exome sequencing could be useful in the diagnosis of unusual clinical features.

## 1. Introduction

Retinal dystrophies (RD) are a group of inherited degenerative disorders characterized by damage to the photoreceptor cells in the retina [1]. The large clinical heterogeneity of RD may be misleading due to the presence of macular edema (ME), the lack of bone-spicule pigmentation, breakdown of the blood–retinal barrier, and a dystrophic sheathing of the retinal vessels [2,3]. All modes of transmission may be involved. However, many cases are sporadic without any family history. Many causal genes remain to be identified.

Despite the good performance of the next-generation sequencing (NGS) of targeted RD genes (PMID 30718709), whole-exome sequencing (WES) combined with clinical analysis has improved the diagnosis of rare syndromic retinal dystrophies [4,5].

The aim of this study was to describe five unrelated cases that presented with retinal or optic nerve lesions associated with systemic abnormalities, and a mutation was identified in the *Casitas B-lineage lymphoma CBL* gene in all cases.

## 2. Results

Five unrelated patients with a mean age of 13 years at the time of analysis (range of 7–32 years), including four females and one male, presented with retinal lesions associated with a mutation in the *CBL* gene. Four of these patients carried homozygous mutations in hematopoietic tissue, and fibroblasts and other peripheral cells showed heterozygous mutations, suggesting constitutional mutations with possible acquired somatic uniparental isodisomy (cases 1 to 4). Previous genetic analysis identified no variants in genes associated with retinitis pigmentosa and inherited optic neuropathy.

Cases 1 and 2 showed ophthalmological similarities: initial ME, a vascular or papillary leakage on FA, visual field constriction, and a severe bilateral and symmetrical alteration of the retinal function were seen on an ERG. The follow-ups showed lesions similar to rod–cone dystrophy. They also showed systemic similarities, including a history of hepatosplenomegaly since childhood and severe primary infection with Epstein–Barr Virus (EBV). 

Cases 3 and 4 showed damage to the optic nerve head. In case 3, bilateral papilledema was found during the assessments for episodes of impaired VA. In case 4, symptomatic optic atrophy was found after a severe bleeding episode. A history of hepatosplenomegaly was found in both cases.

Case 5 was ophthalmologically different and carried a heterozygous mutation in the CBL gene, associated with a mutation in the IKAROS gene.

### 2.1. Case Presentations

A detailed case presentation of all patients is summarized in Table 1.

#### 2.1.1. Case 1

A 24-year-old female patient, complaining since the age of 17 of a transient decrease in vision in both eyes, experienced episodes of retrobulbar optic neuritis, first diagnosed as multiple sclerosis, in the presence of inflammatory multifocal white matter lesions and lymphocytic meningoencephalitis.

The VA was 20/32 in both eyes, with night blindness. The results of the examination are shown in Table 1. A fundus examination showed bilateral papilledema and peripheral deep round pigmented lesions (Figure 1a,b). An FA showed vascular diffusion at the posterior pole, considered inflammatory, supporting an extended use of immune suppressants (Figure 1c,d). An OCT showed intraretinal cysts (Figure 1e,f). Goldmann VF showed narrowing of the peripheral isopteres. A full-field ERG performed according to the protocol of the International Society for Electrophysiology of Vision (ISCEV) showed the absence of b waves under scotopic conditions, mixed response to the limit of extinction, a and b waves in white that were almost indistinguishable under photopic conditions, as well as flicker stimulation. The mfERG showed the absence of a foveal peak in both eyes, and responses of significantly decreased amplitudes in the central 20° area (Table 1 and Figure 1). Aqueous humor sampling was negative for a Herpes virus poly chain reaction including EBV.

At the age of 30, the VA was 20/125 in the RE and 20/80 in the LE. Fundus examination showed an accentuation of the pigmentary changes in the periphery (Figure 1g,h). The VF and ERG worsened (Figure 1k,l and Table 1). At the systemic level, a history of hepatosplenomegaly since childhood, fibrosing interstitial lung disease due to primary infection with EBV at the age of 19, a nephrotic syndrome, and a right parietal pilocytic astrocytoma were found (Table 1). The diagnostic hypotheses of overload disease (Niemann–Pick disease type B, Gaucher disease) or congenital immune deficiency (autoimmune lymphoproliferative syndrome) were ruled out. The mother and grandmother from the maternal side showed normal ophthalmological examinations. WES identified a mutation in hematopoietic tissue with a loss of heterozygosity in the *CBL* gene: Exon 8 c.1149A > G; p. I383M, known to be related to the RASopathy group of genetic conditions.

Immunosuppressive treatments, including oral prednisone, azathioprine, cyclophosphamide, mycophenolate mofetil, and sirolimus, failed to improve visual acuity and angiographic and campimetric features. Ruxolitinib at a dose of 15 mg twice a day was introduced for RASopathy treatment, and a rapid good response in hepato-splenomegaly, interstitial pneumopathy, and pilocytic astrocytoma, was observed. However, no functional and anatomical ophthalmological improvement was seen after 12 months of treatment. Retinal complications occurred in RE as pre-epithelial choroidal neovascularization treated with anti-VEGF intravitreal injection (Figure 1m). Fibrosis of the vascular lesions occurred within 2 weeks.

At the age of 30, the VA was 20/125 in the RE and 20/80 in the LE. Fundus examination showed an accentuation of the pigmentary changes in the periphery (Figure 1g,h). Autofluorescence photographs showed a few hyperautofluorescent punctuate lesions scattered in the posterior pole. An FA showed unchanged vascular leakage in the posterior pole. A central foveal horizontal OCT scan found the CFT had increased to 394 µm in the RE and 414 µm in the LE, whereas the ellipsoid line was not visible within a thickened RPE, associated with an epiretinal membrane. The VF worsened and became tubular (Figure 1k,l). New full-field ERG and mfERG showed worsened retinal dysfunction with increased photopic damage. Using ruxolitinib for RASopathy treatment, a good systemic response was shown, whereas ophthalmologic complications appeared in the RE as pre-epithelial choroidal neovascularization, as seen in the angio OCT (Figure 1m).

#### 2.1.2. Case 2

A 13-year-old male patient presented with a decrease in vision in both eyes since the age of 6. At inclusion, the best-corrected VA was 20/32 in the RE and 20/63 in the LE. There were no signs of inflammation in the anterior segment. Laser flare meter readings were 18.3 ph/ms in the RE and 12.5 ph/ms in the LE. The Tyndall effect in the anterior vitreous was at two crosses in both eyes. The OCT showed ME (Figure 2a,b). Table 1 shows the examination results that were considered inflammatory and supported an extended use of immune suppressants. Five years later, the VA was 20/25 in the RE and 20/40 in the LE with tubular narrowing of the VF. A fundus examination showed peripheral pigmentary changes; small vessels, some of which were empty at the periphery; and a pale, non-edematous papilla, whereas autofluorescence (Figure 2g,h) showed a hyperfluorescent perifoveal ring, retinal angiography (FA in Figure 2i,j) showed unchanged pole posterior leakage despite immunosuppressive treatment, OCT (Figure 2m,n) showed persistent cysts mainly located in the internal nuclear layer, VF (Figure 2o,p) showed narrowing isopteres, and ERG showed results compatible with retinal dystrophy. A full-field ERG showed an absence of responses under scotopic conditions and the presence of a and b waves during the mixed response but their amplitude was extremely reduced, similar to a and b waves in white under photopic conditions, as well as flicker stimulation. On the mfERG, the macular responses were present but hypovolted, the damage being more pronounced in the LE.

At the systemic level, a history of severe EBV infection was recorded at the age of 3, with hepatosplenic and peripheral lymph node tumor syndrome and encephalitis that had regressed under corticosteroids. The aqueous humor PCR was negative for EBV. Immunosuppressive treatments included methotrexate, azathioprine, interferon-alpha, and sirolimus. Resorption of the bilateral ME and VA stabilization were observed, whereas VF tubular constriction worsened.

Panel NGS identified a mutation with a loss of heterozygosity in the CBL gene: Exon 8 c.1141T > G; p.C381G.

#### 2.1.3. Case 3

A 25-year-old female patient had been admitted for a sudden visual loss in both eyes. The initial examination did not find any abnormalities, except for bilateral papillary hyperemia at the fundus examination. Brain magnetic resonance imaging (MRI) showed a right Roland lesion in the hyper signal on the FLAIR images and no ventricular abnormalities. At the systemic level, the patient had chronic hepatosplenomegaly. She suffered from progressive left hemiplegia. The brain lesion biopsy showed gliosis and positive infiltration of the CD68 macrophage. No suspected lesion of histiocytic tumors or lymphoma. Cortical demyelination and microorganisms were absent. A wide work showed normal spinal fluid, negative serologies for histoplasmosis, Schistosomiasis, bilharzia, distomatosis, filariasis, leishmania, trichinellosis, cysticercosis, toxocariasis, and toxoplasmosis. The HIV serology and Quantiferon^®^ were negative. Lysinuric protein intolerance, Transaldolase deficiency, Niemann–Pick disease, acid lipase deficiency, and cholesteryl ester overload disease were ruled out. Systemic corticosteroids were associated with a regression of the left hemiparesis. The VA was stabilized at 20/20 in both eyes and the slit-lamp and fundus examinations showed no abnormalities, except for persistent papillary edema. The full-field ERG and mfERG showed normal retinal activity and normal visual evoked potentials.

Hematological investigations revealed polyclonal B-cell lymphocytosis in blood and bone marrow granular and megakaryocytic proliferation without any dysplasia or fibrosis, compatible with a diagnosis of myeloproliferative neoplasm. Six months of monitoring are scheduled. Obstructive diffuse interstitial pneumopathy occurred after splenectomy.

The NGS panel identified a homozygous mutation in the CBL gene: Exon 9 c.1259G > C; p.Arg420Pro, known to be involved in RASopathy disorders. NGS in myeloïd cells showed *CBL* 98% variant allelic frequency without other mutations.

#### 2.1.4. Case 4

A 7-year-old female patient presented with a sudden loss of vision in the RE following a hemorrhagic tonsillectomy requiring a blood transfusion at the age of 5. The VA was without light perception with exotropia in the RE and was 20/20 in the LE. A fundus examination showed optic atrophy in the RE without other retinal abnormalities and was normal in the LE. The FA and macular OCT were normal. The RNFL thickness was 50 µm in the RE and 75 µm in the LE. No abnormality was found on the brain MRI or during the cardiovascular assessment. The ophthalmological examinations remained stable for 5 subsequent years.

A history of hepatosplenomegaly since childhood was recorded without any other medical history. A WES identified a loss of heterozygosity in the CBL gene: Exon 9 c.1259G > A; p.R420Q.

#### 2.1.5. Case 5

A 13-year-old female patient was monitored since the age of 7 for episodes of idiopathic chronic uveitis in both eyes. The examination results are shown in Table 1. Cataract surgery combined with diagnostic vitrectomy was performed in the LE, complicated by the occurrence of an intraoperative retinal detachment and then a total loss of vision in the LE. Bacteriological and virologic vitreous samples were negative, and a cytologic vitreous sample showed an aspect compatible with an epithelioid reaction. Six years later, the VA deteriorated in the RE to 20/125 with the appearance of a temporal retinal detachment related to the VRT (Figure 3). Local corticosteroid injections and systemic immunosuppressive treatments, including prednisone, methotrexate, adalimumab, tocilizumab, interferon-alpha, and golimumab, were ineffective on the condition of the eyes.

At the systemic level, the patient had a history of idiopathic thrombocytopenic purpura. A WES identified a heterozygous mutation in the CBL gene: Exon 9 c.1258C > G; p.R420G, associated with a mutation in the IKAROS gene.

## 3. Discussion

Whole-genome sequencing is increasingly used to diagnose medical conditions of a genetic origin. We present three cases of retinal damage initially interpreted as inflammatory and treated as chronic bilateral uveitis. However, successive treatments with different immunosuppressive molecules did not lead to changes in angiographic features and failed to prevent retinal damage. This has been suggested to be attributed to inherited retinal degeneration associated with mutations in the *CBL* gene identified on whole-exome sequencing. Variants in genes previously associated with inherited retinal degeneration were lacking.

Several ophthalmological elements may have been in favor of primary uveitis, such as chronic ME, fluorescein leakage in retinal angiography, and cells in the vitreous noted at 1 or 2+ on the Sun scale. However, these three elements are compatible with the diagnosis of retinal dystrophy [1,3,6,7]. In RP, cystoid macular edema may occur in 10–50% of patients, showing intraretinal cysts mainly located in the inner nuclear as in our current work [8]. Several mechanisms in RP patients have been suggested, including a breakdown of the blood–retinal barriers, dysfunction of the pumping mechanism in the RPE, and Müller cell edema and dysfunction associated with antiretinal antibodies and vitreous traction for the genesis of ME [6]. In fluorescein angiography, leakage was found mainly in the posterior pole but also the periphery in most eyes with a typical RP [9]. Clinically relevant uveitis was found to be uncommon, with a uveitis prevalence of 0.26% in the RP patient cohort [10].

Moreover, inflammatory components have been found in the aqueous humor and vitreous fluid of RP eyes. Some cytokines and chemokines, in particular, monocyte chemotactic protein MCP1 and placental growth factor (PlGF), IL2, IL6, interleukin-(IL-)8, interferon gamma-induced protein (IP)-10, hepatocyte growth factor (HGF), platelet-derived growth factor AA (PDGF-AA), and matrix metalloproteinase, were found to be higher than in the controls [11]. Moreover, MCP1 and placental growth factor (PlGF), IL2, and IL6 exceeded the levels of serum suggesting intraocular production [12] They could participate in photoreceptor apoptosis via the activation of the microglia. These findings could suggest a sustained chronic inflammatory reaction that may mediate multiple events such as the recruitment of leukocytes and the enhancement of immune responses. However, a high level of immunosuppression obtained by various immunosuppressive regimens that lasted for years failed to prevent photoreceptor damage and the loss of visual field, as seen in the current study.

Choroidal neovascularization type 2 spreading above the RPE is a rare complication of posterior uveitis, especially affecting the outer retina–retinal pigment epithelium–choroid interface. The secondary inflammatory component described in the current work and the outer retina–retinal pigment epithelium–choroid destructive lesions are underlying conditions that may stimulate neovascular growth via pro-inflammatory prostaglandins and VEGF production.

Several authors have reported a loss of heterozygosity of the mutated *CBL* gene found on chromosome 11q in patients with malignant myeloid hemopathies [13]. This homozygous mutation of the *CBL* gene has also been found in children with JMML [14]. The *CBL* gene encodes an E3 ubiquitin ligase that negatively regulates Ras/MAPK, involved in the control of the RAS phosphorylation pathway. Its mutations lead to a constitutional activation of this pathway [15]. The newly described *CBL* mutation-associated syndrome has been recently reported to show germline mutations in the *Casitas B-lineage lymphoma* proto-oncogene that resemble the Noonan syndrome phenotype and predispose a patient to juvenile myelomonocytic leukemia [16,17,18,19]. *CBL* mutation-associated syndrome is characterized by phenotypic heterogeneity and variable developmental, tumor, and functional expressivity. Clinical features overlap with Noonan syndrome and type 1 neurofibromatosis, suggesting the important role of the *CBL* gene in regulating the RAS pathway, which is essential to the orchestration of the developmental program of many species. Constitutional abnormalities in *CBL* mutation-associated syndrome include impaired growth, developmental delays, cryptorchidism, and late-onset vascular disorders such as optic atrophy, arterial hypertension, and acquired cardiomyopathy [19]. The manifestations of *CBL* syndrome related to RASopathies are still poorly understood, especially with regard to the eye condition, for which no damage description has been associated with a mutation of this gene. Optic atrophy and retinal arteritis have been identified and associated with vascular damage (Table 2) [16,17,18,19,20,21,22,23].

On the one hand, *CBL* gene mutations could therefore be explained by a hereditary mechanism due to a mutated allele from a parent and, on the other hand, by a de novo somatically acquired mechanism corresponding to a secondary genetic event. Somatically, the loss of heterozygosity is due to acquired uniparental disomy, i.e., the inheritance of two copies of the *CBL* gene from the mutated parental allele.

The ophthalmological and systemic phenotype exhibited by the fifth patient did not overlap with previously described *CBL* syndrome patients and with other patients in the current study. On the other hand, the fifth patient’s phenotype did not overlap with the previously described IKAROS mutation phenotype, mainly including immune deficiency, autoimmunity, and malignancy. The common variables of immunodeficiency syndrome, that is, repeated bacterial infections and B cell acute lymphoblastic leukemia, are the main features related to the different IKAROS variants. Moreover, the ophthalmological disease did not respond to local and systemic corticosteroid treatment associated with methotrexate, adalimumab, tocilizumab, interferon-alpha, and golimumab. The therapeutic resistance does not argue in favor of a primary autoimmunity dysfunction process. The phenotype may therefore be linked to the combination of *CBL* and IKAROS mutations.

Cadherin is a protein involved in many ocular structures, especially on the RPE [24]. RNA sequencing analysis of patients’ cells revealed an upregulation of E-cadherin [20]. Therefore, a link between the mutation of the *CBL* gene and the overproduction of E-cadherin could be assumed, and deregulation of its production could lead to the eye damage described in our patients. Our team (CB, SM, FRL) is currently investigating the role of E-cadherin within the RPE and its clinical consequences at the Imagine Institute. Further molecular biology studies are still needed.

## 4. Materials and Methods

This longitudinal prospective follow-up study included five patients with a complex medical history, without a family history of RD, and with signs of ocular inflammation resistant to immune suppressants, the evolution of which suggested a dystrophic process. All patients underwent a comprehensive eye examination including best-corrected visual acuity (VA) measured on decimal and Snellen scales, a slit-lamp examination, and a dilated fundus examination. Goldmann visual field (VF) testing was performed by orthoptists in the department. Color fundus photographs, autofluorescence photographs, fluorescein angiography (FA), and indocyanine angiography (ICGA) were performed using HRA2 (Heidelberg Engineering, Heidelberg, Germany), Optos (Optos PLC, Edinburgh, United Kingdom) and CR2 plus (Canon, Tokyo, Japan). Spectral-Domain Optical Coherence Tomography (SD-OCT) was performed using the Spectralis (Heidelberg Engineering, Heidelberg, Germany). The flare was measured using a laser flare cell meter FC-2000^®^ (Kowa, Tokyo, Japan). Full-field electroretinography (ERG) and multifocal ERG (mfERG) were performed in the clinical neurophysiology department of Pitié-Salpêtrière university hospital according to the protocol of the International Society for Electrophysiology of Vision (ISCEV). An in-depth systemic assessment was performed.

### 4.1. Ethics and Consents

Major and minor patients’ parents signed informed consent in accordance with the French bio-ethics law No. 2011-814, decree 2013-527. The study was approved by our local Ethics Committee. The study was conducted in compliance with good clinical practice and followed the tenets of the Declaration of Helsinki.

### 4.2. Molecular Analysis

Whole-exome sequencing (WES) and the search for mutations involved in retinitis pigmentosa (RP) or rod–cone dystrophy (RCD) were performed at the Imagine Institute. Patient DNA was extracted from the peripheral blood using standard extraction methods for genetic analysis. Sanger sequencing or next-generation sequencing (NGS) such as panel NGS or WES were used. Mutations identified by NGS were confirmed by Sanger sequencing. NGS was performed in all patients who did not carry a mutation in the ALPS or CTLA4 genes. Exons were amplified by polymerase chain reaction (PCR) from the genomic DNA according to standard protocols. The PCR products were sequenced in both directions and analyzed with ApE-A plasmid Editor v2.0.45. Illumina-compatible pre-capture barcoded genomic DNA libraries were constructed according to the manufacturer’s protocol (Ovation Ultralow, Nugen Technologies) from 1–3 µg of DNA. Several pre-capture libraries were pooled at equimolar concentrations. The capture process was performed using this pool and biotinylated probes from the SureSelect panel (Agilent Technologies, Santa Clara, CA, USA). Libraries were sequenced with an Illumina HiSeq2500 (Paired-End sequencing 130 × 130 bases, High Throughput Mode). Sequence readings were aligned to the human hg19 reference genome using the Burrows–Wheeler Alignment version 0.6.2.13. The mean coverage depth obtained for each sample was ≥300×, with ≥97% of the panel regions covered at least 15× and ≥90% of the panel regions covered at least 30×. Exome libraries were prepared using the 50 Mb SureSelect Human All Exon kit V3 (Agilent Technologies, Carlsbad, CA, USA). Paired-end 75 + 35 readings were generated using a SOLiD5500XL (Life Technologies) and mapped using LifeScope (Life Technologies). The mean coverage depth obtained for each sample was ≥70×, with ≥83% of the exome covered at least 15×. For each high-throughput sequencing, single nucleotide polymorphisms and indel calling were performed using GATK tools. In-house software (Polyweb) was used to filter the variants.

## 5. Conclusions

The combination of whole-exome sequencing and clinical analysis allows better diagnosis of rare syndromic retinal dystrophies. The systemic damage described and the ocular damage leading to the retinal dystrophy found could be the new characteristics of the ocular phenotype of *CBL* mutation-associated syndrome.

## Figures and Tables

**Figure 1 ijms-23-07868-f001:**
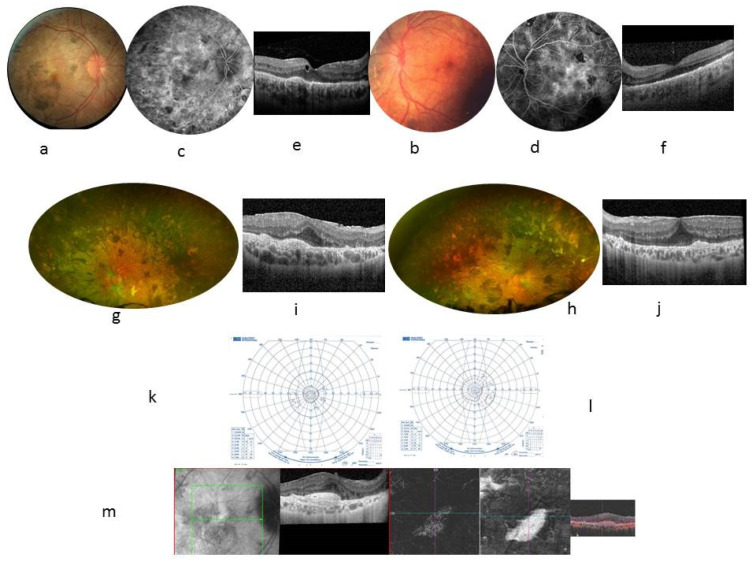
Case 1. 24-year-old woman. BCVA at 20/32 in both eyes. Color fundus photographs showing many pigmentary changes (**a**,**b**), vascular leakage at the posterior pole on FA at 10 min (**c**,**d**), and a macular thickening seen in central foveal horizontal scan (**e**,**f**) showing a central foveal thickness (CFT) at 272 µm in the RE and 259 µm in the LE, with presence of cysts within the inner nuclear layer and external plexiform layer adjacent to the papilledema; the external retinal layers are not visible but the external limiting membrane remains visible in the retrofoveolar area. The mean thickness of the retinal nerve fiber layer (RNFL) was 290 µm in the RE and 333 µm in the LE. At the age of 30, fundus examination showed an accentuation of the pigmentary changes in the periphery (**g**,**h**). Autofluorescence photographs showed a few hyperautofluorescent punctuate lesions scattered in the posterior pole (**i**,**j**). A central foveal horizontal OCT scan found the CFT had increased to 394 µm in the RE and 414 µm in the LE, whereas the ellipsoid line was not visible within a thickened RPE, associated with an epiretinal membrane. The VF worsened and became tubular (**k**,**l**). RE showed pre-epithelial choroidal neovascularization, as seen in the angio OCT ×20 magnification (**m**).

**Figure 2 ijms-23-07868-f002:**
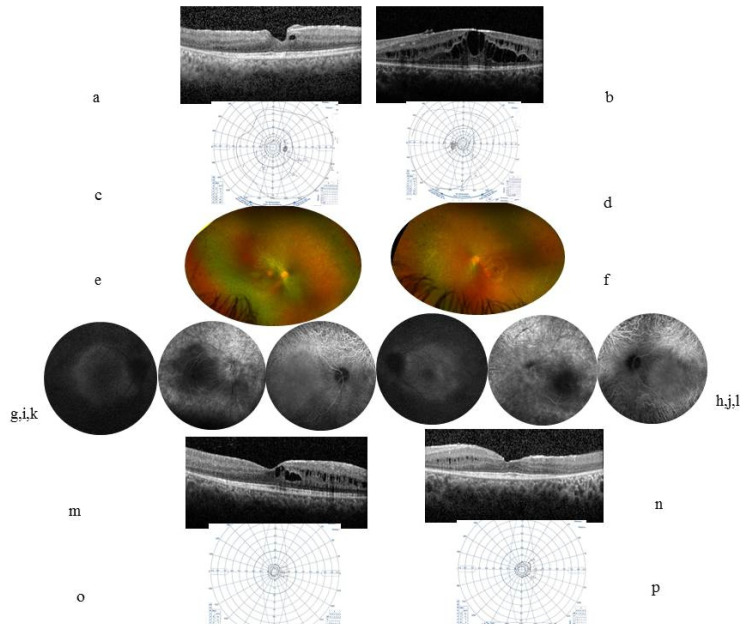
Case 2. At inclusion, a 13-year-old boy’s (**a**) (RE) (**b**) (LE) horizontal central macular OCT showed ME with a CFT at 344 µm in the RE and 467 µm in the LE. (**c**) (RE) (**d**) (LE) VF Goldmann was relatively preserved. Five years later, the VA was 20/25 in the RE and 20/40 in the LE. (**e**) (RE) (**f**) (LE) color fundus photographs showed small vessels and a pale papilla and a whitish annular scale bar ×10 perifoveal ring that corresponded to the hyperautofluorescent annular ring in autofluorescence pictures (**g**,**h**). (**i**) (RE) (**j**) (LE) FA showed macular and papillary leakage. (**k**) (RE) (**l**) (LE) ICGA showed visible choroidal vessels in the periphery. (**m**) (RE) (**n**) (LE) OCT showed persistent cysts that are mainly located within the internal nuclear layer and disappearance of the external layers including the ellipsoid and external nuclear layers at 1000 µm from the center; the RNFL thickness was stable, at 95 µm in the RE and 109 µm in the LE. (**o**) (RE) (**p**) (LE) tubular narrowing VF.

**Figure 3 ijms-23-07868-f003:**
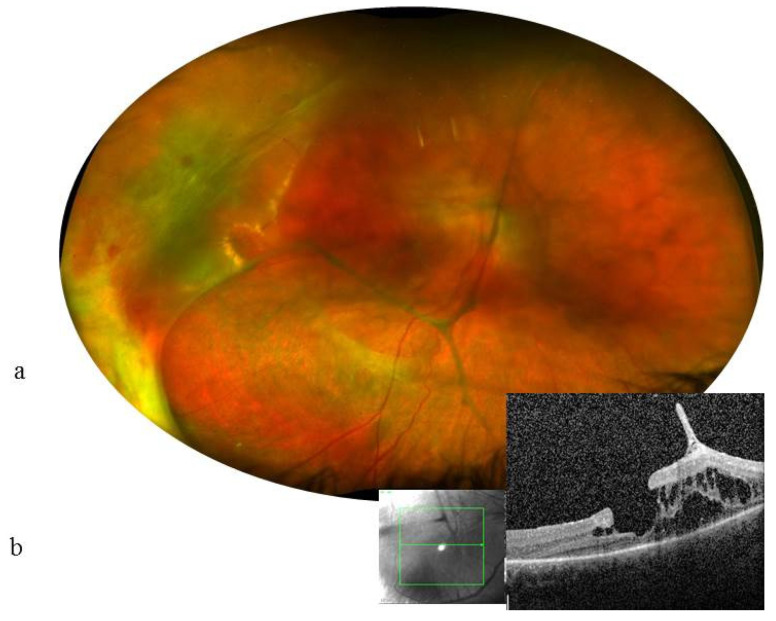
Case 5. 13-year-old girl. VA at 1.6/10 (0.8 LogMAR) in the RE and no light perception in the LE. RE: (**a**): Color fundus photograph shows many thick condensations of the posterior hyaloid and a detachment of the temporal retina. (**b**) horizontal scan (green lines) OCT showed intraretinal cysts, vitreoretinal traction, and a lamellar hole in the RE (magnification ×20). Cysts are mainly located within the internal nuclear layer and the external retinal layers; the external nuclear, external limiting, and ellipsoid layers have disappeared at 1000 µm from the center.

**Table 1 ijms-23-07868-t001:** Summary of clinical features and management of CBL patients.

Patients at Admission	BCVA Snellen RE LE	Aqueous Humor Cells RE LE	Flare Ph/ ms RE LE	Vitreous Cells RE LE	Papillary Edema RE LE	Fundus Periphery in Both Eyes	Retinal Vascular Leakage	CFT Microns RE/LE// Intra Retinal Cysts	VF Goldmann in Both Eyes	ffERG mfERG	Hepato-Splenomegaly	Interstitial Pneumopathy	History of Severe EBV Infection	Brain Lesion	*CBL* Variant	Immunosuppressive Treatment Efficiency
F. 24	20/32 20/32	0.5+ 0.5+	3 2	1+ 1+	+ +	Round pigment mottling	yes	272/259 Intraretinal cysts present	Isopteres narrowing	absence of b waves under scotopic conditionsa bsence of foveal peak	Since childhood	present	+	Pilocytic astrocytoma	Exon 8 c.1149A > G; p. I383M	No Ruxolitinib = no eye efficiency resolved splenomegaly, pneumopathy and pilocytic astrocytoma
M. 13	20/32 20/63	1+ 1+	18 12	2+ 2+	+ +	Round pigment mottling	yes	344/467 Intra retinal cysts present	Normal at admission	Absence of reponses in scotopic conditions Hypovolted foveal pick	Since childhood	absent	+	No lesion	Exon 8 c.1141T > G; p.C381G.	No efficiency
F. 25	20/20 RLE	0.5+ 0.5+	nd	0.5+ 0.5+	+ +	normal	no	No intra retinal cysts	Slight enlarged blind spot	Normal	Since childhood Myelo Prolife rative sd	present with obstructive sd	-	Roland lesion	Exon 9 c.1259G > C; p.R420P	NA
F 7	NoLP 20/20	0.5+ 0.5+	nd	0.5+ 0.5+	papillary atrophy in RE	normal	no	205/211 No intra retinal cysts	No recordable in RE Normal in LE	nd	Since childhood	absent	-	No lesion	Exon 9 c.1259G > A; p.R420Q.	no efficiency
F 13	20/100 20/200	1+ 1+	24 14	2+ 2+	+ +	Pigment mottling	yes	236/208 Vitreoretinal Traction and Intraretinal cysts in both eyes. RE lamellar hole	nd	nd	absent	absent	-	No lesion	Exon 9 c.1258C > G; p.R420G, associated with IKAROS gene variant	no efficiency

**Table 2 ijms-23-07868-t002:** Summary table of studies included in CBL.

Ophthalmological Features in *CBL* Mutation-Associated Syndrome References	Number of Patients	Macroscopic Ophthalmological Features Number of Patients	Intra-Ocular Features Number of Patients
C. Niemeyer et al., 2010 [16]	21	Noonan and NF1 sd *n* = 21	Optic atrophy *n* = 4/21 Takayasu arteritis *n* = 1/21
S. Martinelli et al., 2015 [19]	5	Noonan-like sd *n* = 5	0
Strullu et al., 2013 [20]	5	Noonan-like sd *n* = 5	0
Bülow et al., 2015 [21]	3	Hypertelorism Mild ptosis Downslanting palpebral fissures *n* = 2/3	0
Perez et al., 2010 [18]	4	Facial dysmorphism including hypertelorism epicanthic folds *n* = 3/4	0
Hyakuna et al., 2015 [22]	1	Hypertelorism edematous eye lids	0
Becker et al., 2014 [23]	1	0	0
Our current study	5	Hypertelorism Downslanting palpebral fissures *n* = 3	Pseudo uveitis *n* = 3/5 Optic atrophy *n* = 1/5 Papillar edem *n* = 4/5

NF1 = neurofibromatosis type 1.

## Data Availability

The authors have full access to the presented data. Data are available on request due to restrictions e.g., privacy or ethical. The data presented in this study are available on request from the corresponding author.

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
