# Peer review of "Casitas B-lineage lymphoma Gene Mutation Ocular Phenotype"

_ijms, 2022, doi:10.3390/ijms23147868_

Round 1
Reviewer 1 Report
This study present case details on five unrelated patients (4 females and 1 male), who all have mutations in the CBL gene (one also has a heterozygous mutation in IKAROS). Usefully, none of these patients had variants in known genes associated with RP or optic neuropathy.
The diagnostic manifestations of each patient are well-described with illustrative figures highlighting the ocular pathologies seen. Assays performed include fundus examination, OCT, visual field testing, ERG, and fluorescein angiography. The similarities and differences in presentation between patients are clear and well laid out. There are some inconsistencies in descripition of the CBL mutations - the phrase "loss of heterozygosity" is used to describe WES outcomes, but this suggests that there has already been a loss (or mutation) in one copy of CBL, and that another mutation is subsequently found. The authos should consider describing the status of both CBL alleles in the described patients. In the case of patient 3, for example, the mutation is described as homozygous. Related, patient 5 has a heterozygous mutation in CBL as well as a hetoerzygous mutation in IKAROS - this genetic interaction should be discussed.
That authors have carried out a nice study with good case details provided. Furthermore, the brief discussion of the difficulties associated with diagnosis of ocular diseases that the authors provide is relevant and useful. Similarly, the paragraphs describing the role of CBL, and its potential role in eye disease is good. Finally, the observation that E-cadherin is upregulated in these patients provides insight into both potential disease mechanisms and may be used to design basic science experiments to investigate the mechanism further.
Given the interesting ophthalmological findings in patient 5, who has mutations in both CBL and IKAROS, a discussion of the potential interaction between these two factors might be good. However, it may be that space constraints do not permit expansion on this topic.
Minor corrections:
Should be reviewed with English language assistance. e.g. line 24 does not make sense "two cases sharing ophthalmological as 24
chronic macular edema"
For example, line 177 "Glodmann" should be "Goldmann". There are several others.
Not all figure subpanels are referred to in the text.
Author Response
RESPONSES TO REVIEWER 1
Thank you for your review ; we have taken in account all the remarks. We have written in color ink the changes in the manuscript.
This study present case details on five unrelated patients (4 females and 1 male), who all have mutations in the CBL gene (one also has a heterozygous mutation in IKAROS). Usefully, none of these patients had variants in known genes associated with RP or optic neuropathy.
The diagnostic manifestations of each patient are well-described with illustrative figures highlighting the ocular pathologies seen. Assays performed include fundus examination, OCT, visual field testing, ERG, and fluorescein angiography. The similarities and differences in presentation between patients are clear and well laid out. There are some inconsistencies in descripition of the CBL mutations - the phrase "loss of heterozygosity" is used to describe WES outcomes, but this suggests that there has already been a loss (or mutation) in one copy of CBL, and that another mutation is subsequently found.
RESPONSE : All patients except patient 5 have a pathogenic variant in CBL gene with an homozygous state in hematopoetic cells while peripheral fibroblast cells showed the CBL variant in heterozygositous status. The constitutional mutation in the hematopoetic tissu has been suggested to be the result of aquired uniparental disomy. We therefore deleted « loss of heterozygosity ». We added these data in the results section.
Related, patient 5 has a heterozygous mutation in CBL as well as a hetoerzygous mutation in IKAROS - this genetic interaction should be discussed.
RESPONSE : On the one hand, ophthalmological and systemic phenotype exhibited by the 5th patient did not overlap with previously described CBL syndrome patients and with others patients of the current study. In the other hand the 5th patient’s phenotype did not overlap with previously described IKAROS mutations phenotype, including mainly immune deficiency, autoimmunity and malignancy . The common variable immunodeficiency syndrome, repeated bacterial infections, and B cell acute lymphoblastic leukemia are the main features related to different IKAROS variants. Moreover the ophthalmological disease did not respond to local and systemic corticoseroid treatment associated with methotrexate, adalimumab, tocilizumab, interferon alpha, and golimumab. The therapeutic resistance does not argue in favour of a primary auto-immunity dysfunction process. The phenotype may therefore be linked to the combination of CBL and IKAROS mutations. We added this paragraph in the discussion.
That authors have carried out a nice study with good case details provided. Furthermore, the brief discussion of the difficulties associated with diagnosis of ocular diseases that the authors provide is relevant and useful. Similarly, the paragraphs describing the role of CBL, and its potential role in eye disease is good. Finally, the observation that E-cadherin is upregulated in these patients provides insight into both potential disease mechanisms and may be used to design basic science experiments to investigate the mechanism further.
Given the interesting ophthalmological findings in patient 5, who has mutations in both CBL and IKAROS, a discussion of the potential interaction between these two factors might be good. However, it may be that space constraints do not permit expansion on this topic.
Minor corrections:
Should be reviewed with English language assistance. e.g. line 24 does not make sense "two cases sharing ophthalmological as 24 chronic macular edema"
RESPONSE : The sentence has been corrected. The results included two cases sharing ophthalmological signs including …
For example, line 177 "Glodmann" should be "Goldmann". There are several others.
RESPONSE : Goldmann and others have been corrected .
Not all figure subpanels are referred to in the text.
RESPONSE : references to figure subpanels have been added in the text.
Reviewer 2 Report
1. The references in line 395-422 is unreadable. Needs to be corrected.
2. Line 24-25: The authors should give more description in the methods and results when describing electroretinography alteration. Readers can read the detailed methods but it would be easier to read if this info was provided in the description of the experimental results.
3. How about the choroidal neovascularization? Please describe it in Discussion, even if it is an excerpt from the reported cases.
4. FIG1, 2, and 3. The authors should provide representative figures showing the labelling they produced.
5. Table 1. Summary of clinical features and management of CBL patients. Needs to be improved.
Author Response
RESPONSES TO REVIEWER 2
Thank you for your review ; we have taken in account all the remarks. We have written in color ink the changes in the manuscript.
- The references in line 395-422 is unreadable. Needs to be corrected.
RESPONSE : wrong references has been corrected
- Line 24-25: The authors should give more description in the methods and results when describing electroretinography alteration. Readers can read the detailed methods but it would be easier to read if this info was provided in the description of the experimental results.
RESPONSE : Methods used for electroretinography has been added in the description of the experimental results.
- How about the choroidal neovascularization? Please describe it in Discussion, even if it is an excerpt from the reported cases.
RESPONSE : the choroidal neovascularisation type 2 spreading above the RPE is a rare complication of posterior uveitis, specially affecting the outer retina-retinal pigment epithelium-choroid interface. The inflammatory secondary component described in the current work and the outer retina-retinal pigment epithelium-choroid destructive lesions are underlying conditions that may stimulate neovascular growth. This has been added in the discussion section. (references : Mansour AM, Arevalo JF, Fardeau C, Hrisomalos EN, Chan WM, Lai TY, Ziemssen F, Ness T, Sibai AM, Mackensen F, Wolf A, Hrisomalos N, Heiligenhaus A, Spital G, Jo Y, Gomi F, Ikuno Y, Akesbi J, LeHoang P, Adan A, Mahendradas P, Khairallah M, Guthoff R, Ghandour B, Küçükerdönmez C, Kurup SK. Three-year visual and anatomic results of administrating intravitreal bevacizumab in inflammatory ocular neovascularization. Can J Ophthalmol. 2012 Jun;47(3):269-74. // Mansour AM, Arevalo JF, Ziemssen F, Mehio-Sibai A, Mackensen F, Adan A, Chan WM, Ness T, Banker AS, Dodwell D, Chau Tran TH, Fardeau C, Lehoang P, Mahendradas P, Berrocal M, Tabbarah Z, Hrisomalos N, Hrisomalos F, Al-Salem K, Guthoff R. Long-term visual outcomes of intravitreal bevacizumab in inflammatory ocular neovascularization. Am J Ophthalmol. 2009 Aug;148(2):310-316.)
- FIG1, 2, and 3. The authors should provide representative figures showing the labelling they produced.
RESPONSE : Figures labelling has been corrected
- Table 1. Summary of clinical features and management of CBL patients. Needs to be improved.
RESPONSE : Table 1 has been improved